# Reconstructing Depth Images for Time-of-Flight Cameras Based on Second-Order Correlation Functions

Tian-Long Wang [1], Lin Ao [2], Jie Zheng [2] and Zhi-Bin Sun [1,3,*]

1    National Space Science Center, Chinese Academy of Sciences, Beijing 100190, China; wangtianlong0726@163.com
2    Northeastern University, Shenyang 110819, China; aolin0102@163.com (L.A.); 2310683@stu.neu.edu.cn (J.Z.)
3    University of Chinese Academy of Sciences, Beijing 100049, China
*    Correspondence: zbsun@nssc.ac.cn

**Abstract:** Depth cameras are closely related to our daily lives and have been widely used in fields such as machine vision, autonomous driving, and virtual reality. Despite their diverse applications, depth cameras still encounter challenges like multi-path interference and mixed pixels. Compared to traditional sensors, depth cameras have lower resolution and a lower signal-to-noise ratio. Moreover, when used in environments with scattering media, object information scatters multiple times, making it difficult for time-of-flight (ToF) cameras to obtain effective object data. To tackle these issues, we propose a solution that combines ToF cameras with second-order correlation transform theory. In this article, we explore the utilization of ToF camera depth information within a computational correlated imaging system under ambient light conditions. We integrate compressed sensing and non-training neural networks with ToF technology to reconstruct depth images from a series of measurements at a low sampling rate. The research indicates that by leveraging the depth data collected by the camera, we can recover negative depth images. We analyzed and addressed the reasons behind the generation of negative depth images. Additionally, under undersampling conditions, the use of reconstruction algorithms results in a higher peak signal-to-noise ratio compared to images obtained from the original camera. The results demonstrate that the introduced second-order correlation transformation can effectively reduce noise originating from the ToF camera itself and direct ambient light, thereby enabling the use of ToF cameras in complex environments such as scattering media.

**Keywords:** time-of-flight; computational correlation imaging; scattering media; compressed sensing; untrained neural network

## 1. Introduction

With the development of technology, 3D imaging has been applied in many fields, and depth images have become one of the most important and innovative areas in the field of image sensing science and engineering in recent decades [1]. In applications such as object recognition and remote sensing, depth maps are important tools for capturing the spatial position and motion of detected objects. In the past few decades, 3D imaging technology has mainly been divided into three categories, structured light 3D imaging [2], binocular vision 3D imaging [3–6] and time-of-flight (ToF) 3D imaging [7–9]. The basic principle of structured light 3D imaging is optical triangulation, which reconstructs depth maps by geometric distortions caused by the shape of the object's surface. Binocular vision 3D imaging uses two eyes (or two images) to capture the shape and position of objects in the scene, calculating 3D information including depth and size by detecting feature points of the objects and the disparity between the two images. Compared to structured light 3D imaging and binocular vision 3D imaging, ToF technology is a 3D imaging technology that uses the time difference of light beam reflection on the object's surface to measure distance. ToF imaging technology is largely dominated by ToF cameras, as they have small sizes, good resolution, robustness in ambient light conditions, low power consumption, and

fast processing capabilities. ToF cameras have advantages in distance measurement compared to traditional cameras because they can measure without being limited by lighting conditions. ToF cameras typically operate within a range of tens of meters to capture high-resolution 3D images at video rates. ToF cameras use modulated infrared light sources to illuminate the scene and resolve the phase shift distance measured by the sensor. However, the quality of depth maps obtained by ToF cameras is easily affected by ambient light and limitations of the camera itself, and phase shift reconstruction is sensitive to noise. In the past few decades, pattern modulation and bucket detection techniques have been proven effective in 3D imaging. Howland et al. achieved single-pixel 3D imaging of the region of interest using distance gating technology and obtained a complete depth map through full-range scanning [10]. Kirmani et al. used a time-correlated single-photon counting system instead of distance gating to obtain depth maps [11]. Sun Mingjie et al. simplified Kirmani's structure and determined the depth map by sampling the time-varying intensity measured by a high-speed photodiode [9]. Sun Baoqing et al. used a spatially separated single-pixel detector to measure different shadow images and reconstructed 3D images through multi-view stereo vision [12]. Computational ToF imaging has been proven to be a new method for solving many problems in ToF imaging and promoting new applications, such as transient imaging [13–15], non-line-of-sight imaging [16–18], light transport analysis [19], lensless imaging [20], distance and velocity synchronized imaging, etc. [21]. These methods are mostly based on the waveform design ideas of time-of-flight sensors and temporary modulation codes of light sources. However, though great progress has been made for ToF cameras, the low pixel resolution and signal-to-noise ratio (SNR) in these applications are still bottlenecks that have not been broken through so far [22]. More than a decade ago, the theoretical concept of Computational Ghost Imaging (CGI) was introduced [23–25]. It involves the capture of single-pixel bucket-detector measurements on a detection plane, with subsequent data processing to retrieve the intensity distribution of a reference beam. The fluctuation correlation between the signals acquired by the bucket detectors and the reference signal is utilized to reconstruct the image of the target object. This can be achieved even in challenging environments, such as atmospheric turbulence [26] or scattering media [27], where traditional imaging methods may prove ineffective. Subsequently, the success of Compressed Sensing (CS) algorithms [28–31] in CGI has been observed. CS algorithms are well-suited for the reconstruction of sparse targets. Given the sparsity and noise present in the depth images acquired by ToF sensors, we have opted to employ CS algorithms for processing these images, as they offer robust denoising capabilities and high resolution at low sampling rates. While CS has improved the performance of CGI through its image reconstruction algorithms, its application is constrained by strong sparsity assumptions and limitations in the reconstruction process [32–34]. The emergence of Deep Learning (DL) [35–39] presents an opportunity to relax sparsity constraints by recovering images at ultra-low sampling rates (SR) using trained data and untrained strategies [40–43]. Concurrently, ToF-based novel computational 3D imaging techniques show great promise across various imaging domains [44–47].

In this paper, a time-modulated randomized spatial illumination pattern and a ToF sensor are used to acquire depth maps. The correlation image sensor does not obtain the phase of the measurement pattern directly, but by obtaining the correlation between the received and reference signals. In order to reduce the noise, a second-order correlation transform is introduced to reconstruct the depth map by combining the ToF principle with CGI, CS, and untrained neural networks. It is found that comparing the depth maps from the original ToF camera, the scheme based on compressed sensing and untrained neural network can obtain a higher peak signal-to-noise ratio (PSNR). Specifically, the TVAL3 algorithm and untrained neural networks yield superior reconstruction images characterized by enhanced image quality, heightened SNR, and increased contrast at a 12.5% sampling rate. In addition, based on the ToF camera, we also conducted experiments with the ToF camera through scattering media, in which the CGI-based CS and DL methods can clearly recover the images, while the ordinary ToF method cannot. We believe that the

ToF camera can be used in similar applications where it can be applied to quite complex environments such as scattering media or underwater.

## 2. Basic Theories and Principles

### 2.1. Ranging Principle of ToF Cameras

We start from the continuous-wave (CW) modulation of the illuminating source to introduce the ranging principle of the ToF camera as an example. Generally, a ToF camera consists of two parts, i.e., the illuminating source and the detection system. In use, a scene is illuminated by the modulated light source and the reflected beam that carry the information of the scene observed by the sensor in the detection system, where the phase shift between the light source and the reflected light can be measured and then directly converted into the distance between the scene and the optical source according to the ranging formula Equation (1).

The ranging principle of the CW modulated ToF is shown in Figure 1. For simplicity, we take a square wave modulation of the illuminating optical source as an example to illustrate the basic theory of the ToF. The phase delay $\varphi$ between the modulated emitting and the reflected square wave signals is assumed to be as just shown on the top of Figure 1, which is measured by using the well-known four-step phase shift method [48–50]. Whereas, if $\varphi$ is measured, the distance from the target object to the ToF camera can be calculated according to the following formula [51,52].

$$d = \frac{c}{2} \frac{\varphi}{2\pi f}, \tag{1}$$

where $c$ is the speed of light in a vacuum and $f$ is the signal frequency. It is shown in Figure 1 that the four-phase control signals $C_1 - C_4$ have $\frac{\pi}{2}$ phase delays from each other, which determine the electric charge values $Q_1 - Q_4$ that are the amount of electric charge for the control signals. Generally, the $\varphi$ can be estimated by using $Q_1 - Q_4$ like

$$\varphi = \tan^{-1}\left(\frac{Q_4 - Q_3}{Q_1 - Q_2}\right). \tag{2}$$

However, $Q_1 - Q_4$ are not able to be directly measured, and the modulated signal is not always a square wave. For a generic output signal $c(t)$ of a ToF detector, it can be expressed by the cross-correlation of the emitted signal $g(t)$ and the received signal $s(t)$ as

$$c(\tau) = s(t) * g(t) = \lim_{T \to \infty} \frac{1}{T} \int_{-\frac{T}{2}}^{\frac{T}{2}} s(t)g(t + \tau)dt, \tag{3}$$

where

$$s(t) = 1 + a\cos(\omega t - \varphi), \tag{4}$$

and

$$g(t) = \cos(\omega t), \tag{5}$$

Here, $a$ is the attenuated amplitude of the reflected light signal acquired by the sensor and $*$ indicates convolution.

After the calculation, we can know

$$\begin{aligned} c(\tau) &= \lim_{T \to \infty} \frac{1}{T} \int_{-\frac{T}{2}}^{\frac{T}{2}} [1 + a\cos(\omega t - \varphi)] \cdot [\cos(\omega t + \omega \tau)]dt, \\ &= \frac{a}{2} \cdot \cos(\varphi + \omega \tau), \end{aligned} \tag{6}$$

where $c(\tau)$ represents the ToF sensor's output during period $T$. It actually amounts to the total number of photons received in a period by the certain pixel of the ToF sensor, which is in direct proportion to the electric charge values accumulated on this pixel that are

mentioned in Equation (2). Therefore, $c(\tau_1) - c(\tau_4)$ can usually be substituted for $Q_1 - Q_4$ in Equation (2), where $\tau_1 - \tau_4$ are selected as $0$, $\frac{\pi}{2}$, $\pi$ and $\frac{3\pi}{2}$, respectively. The phase shift $\varphi$, i.e., Equation (2), is rewritten accordingly as

$$\varphi = \tan^{-1}\left(\frac{c(\tau_4) - c(\tau_3)}{c(\tau_1) - c(\tau_2)}\right), \tag{7}$$

while the amplitude $a$ is also computed by the following equation:

$$a = \frac{1}{2}\sqrt{[c(\tau_4) - c(\tau_3)]^2 + [c(\tau_1) - c(\tau_2)]^2}. \tag{8}$$

Eventually, it is obvious that the distance $d$ can be estimated pixel by pixel via substituting $\varphi$ in Equation (7) into the Equation (1).

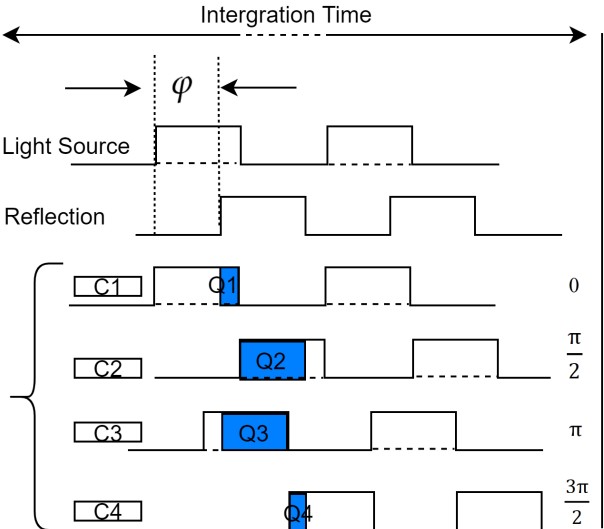

**Figure 1.** The ranging principle of the CW modulated ToF.

### 2.2. CGI via a ToF Camera

A classical passive CGI scheme is shown in Figure 2, where a target object illuminated by a light beam is imaged on a DMD and then one of the reflected beams from the DMD is captured by a single pixel bucket detector through a collecting lens. As an indirect imaging method, the image can be retrieved by the correlation computation between the modulated matrix and the actually captured single-pixel optical signals that carry the information of the target object, which can be expressed by the normalized intensity correlation as

$$g^{(2)}(x,y) = \frac{(1/N)\sum\limits_{i=1}^{N} S_i \phi_i(x,y)}{(1/N)\sum\limits_{i=1}^{N} S_i (1/N)\sum\limits_{i=1}^{N} \phi_i(x,y)}, \tag{9}$$

Here, $S_i$ and $\phi_i(x,y)$ are the $i$th ($i = 1, 2, \ldots, N$) single-pixel signals and the $i$th modulation speckles, respectively. $x$ and $y$ ($x, y = 1, 2, \ldots, M$) are the row and column pixel coordinates of each modulated basis ($M \times M$) of the DMD.

It is well-known that a ToF sensor can export both the intensity and depth data that carry the information of the target object simultaneously, where both of them come from the counting number of photons received by the ToF sensor in a certain period. Differently, the intensity maps can be acquired directly as a universal detector does while the depth maps are exported by counting the number of photons of different phase signals through a complex mathematical operations mentioned in Equation (1). However, they are essentially determined by the statistics of photon numbers. Therefore, both of them can be available in

a CGI system when the ToF camera is used as a bucket detector. That is to say, if the bucket signal $S_i$ is replaced by $D_i$, the Equation (9) still works. Therefore, it would be expressed by

$$g_d^{(2)}(x,y) = \frac{(1/N)\sum\limits_{i=1}^{N} D_i\phi_i(x,y)}{(1/N)\sum\limits_{i=1}^{N} D_i(1/N)\sum\limits_{i=1}^{N} \phi_i(x,y)}, \tag{10}$$

where $D_i = \sum\limits_{x',y'=1}^{P_1,P_2} d_i(x',y')$, which is the $i$th ($i = 1, 2, \ldots, N$) single-pixel signal of the ToF depth maps $d_i(x',y')$. The size of a ToF sensor is assumed to be $P_1 \times P_2$ pixels and its row and column pixel coordinates are $x'(x' = 1, 2, \ldots, P_1)$ and $y'(y' = 1, 2, \ldots, P_2)$, respectively.

Although both the intensity and depth maps can be directly extracted from the raw data exported by the ToF camera, they have different map formats due to their different data acquisition modes. For the intensity maps, the 8-bit gray scale format ranging from 0 to 255 is one of the most-used picture formats. The value in each pixel of each map is positively correlated with the photon number received in this pixel of the ToF sensor. As for the depth maps, the value in each pixel is the distance between a certain point on the surface of the target object and the ToF sensor, which ranges from 1 to 5 m with a step of 0.25 m for the ToF camera we used in the experiments. The values that are less than 1 or larger than 5 would be automatically set to be 1 or 5 by the camera. In addition, the values of those pixels without receiving any photon are also set to be 5. Therefore, the depth maps are generally so-called negative images, where the background has bigger pixel values than those of the image of the target in a depth map. It will be seen in the following that the images reconstructed by the depth maps are also negative ones regardless of the CGI and CS algorithms. For convenience, we call a reconstructed image by utilizing the intensity or depth maps of a ToF sensor an intensity or depth image.

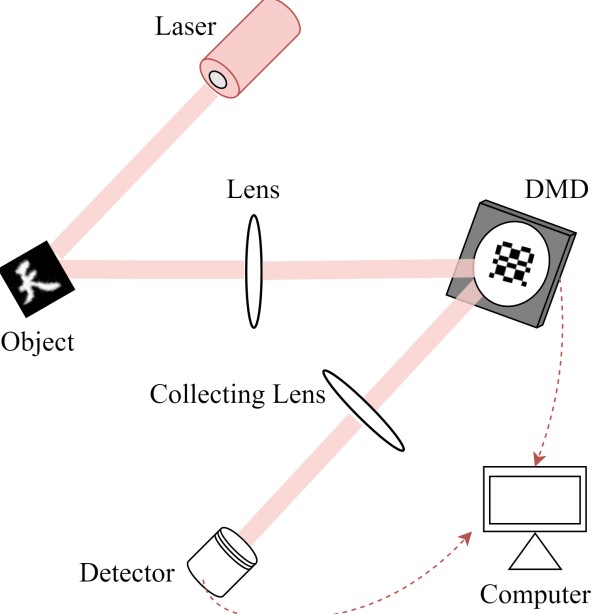

**Figure 2.** The schematic diagrams of CGI. The size of the object is 32 × 32 pixels, the focal length of the converging lens is 50mm, the detector has a resolution of 1280 × 960 pixels, and the dashed line connecting to the computer represents the computer simultaneously controlling the detector and DMD.

### 2.3. Image Reconstruction Algorithm Based on the U-Net Framework

With the development of DL theory, it has been widely used to solve various inverse problems in computational imaging. However, traditional data-driven DL methods require a large amount of training data to optimize network parameters, and the training time is as long as several hours or even days, which hinders its practical application. In recent years, untrained DL methods that combine DIP theory with physical imaging processes have attracted a lot of attention in computational imaging. The DIP theory can be combined with the physical process of single pixel imaging to form a network framework that can be iterated continuously [53–56].

In Figure 3, the U-net framework requires an image of the same size as the modulated speckle as the neural network input, and the U-net network will output the estimated image based on the current weight proportion in the network. In the physical process of traditional single pixel imaging, after ignoring the influence of background noise, the obtained bucket signal can be simply described by the following formula:

$$y = HO^{\mathrm{T}}, \tag{11}$$

where $y$ is the light intensity value of the barrel signal obtained by the single pixel detector, $H$ is the 2D matrix of the modulation matrix speckles, and $O^{\mathrm{T}}$ can be regarded as the 2D reflected light (transmitted light) intensity of the object.

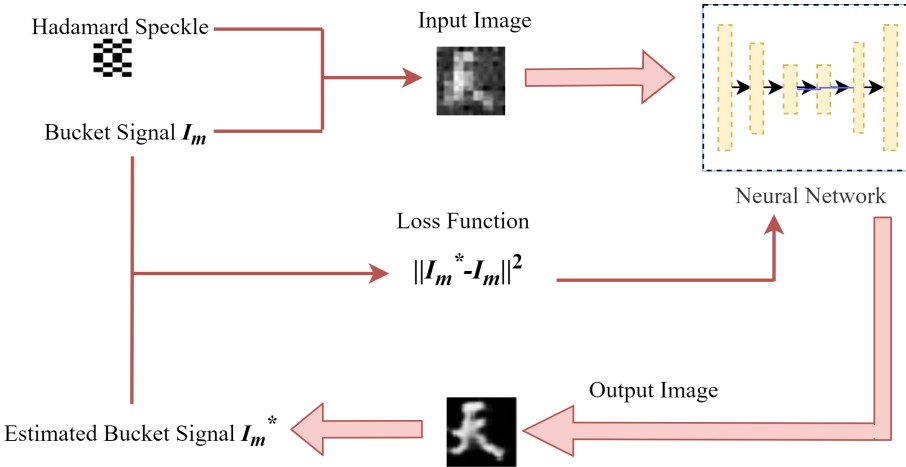

**Figure 3.** Schematic diagram of the image reconstruction using a neural network.

In this neural network framework, the corresponding predicted bucket signals can be obtained based on the predicted images in the network. However, since the output images in the early iterations are of low quality, there are apparent differences between the predicted bucket signals and the real bucket signals. These differences can be regarded as a minimization problem. We take the mean square error (MSE) between the predicted bucket signal and the real bucket signal as the loss function in the neural network. The weights in the neural network are continuously updated by several iterations to minimize the loss function so that the network can output the predicted picture that approximates the real object image. Therefore, in the framework of this neural network, the reconstruction formula of the target object can be represented by the following function:

$$\Re_{\theta^*} = \arg\min_{\theta} \left\| H\Re_{\theta}\left(O_{input}\right) - y'^t \right\|^2 + \xi T\left[\Re_{\theta}O_{input}\right]. \tag{12}$$

Therefore, this neural network framework is different from the traditional neural network methods [57,58] for it does not require training data sets in the process of reconstructing images, and it only needs the bucket signals obtained by the ToF camera and the estimated bucket signal output by the network.

## 3. Experimental Scheme and Analysis of Results

To verify the feasibility of the proposed scheme, a passive CGI experiment with the aid of a ToF camera was performed based on the experimental setup in Figure 2. This setup was similar to the traditional CGI, as depicted in Figure 4, but uses a 320 × 240 pixels ToF camera (OPT8241, Texas Instruments, Dallas, TX, USA) instead of a SPD. Additionally, a total reflection prism (TRP) was inserted to adjust optical paths. In Figure 4, a printed binary object illuminated by an optical beam with a wavelength of 850 nm from an infrared light source is imaged by an imaging lens with a focal length of 50 mm on the surface of the optical unit of a DMD (DLP LightCrafter 4500, Texas Instruments), where the TRP is inserted in the optical path to be suitable for the adjustment of the detection system. One of the reflected beams by the DMD, which carries the information of encoded patterns, was captured by the ToF camera that was synchronized with the light source of the camera at the time of delivery. Both the intensity and depth data exported by the camera are obtained by being computed inside the camera.

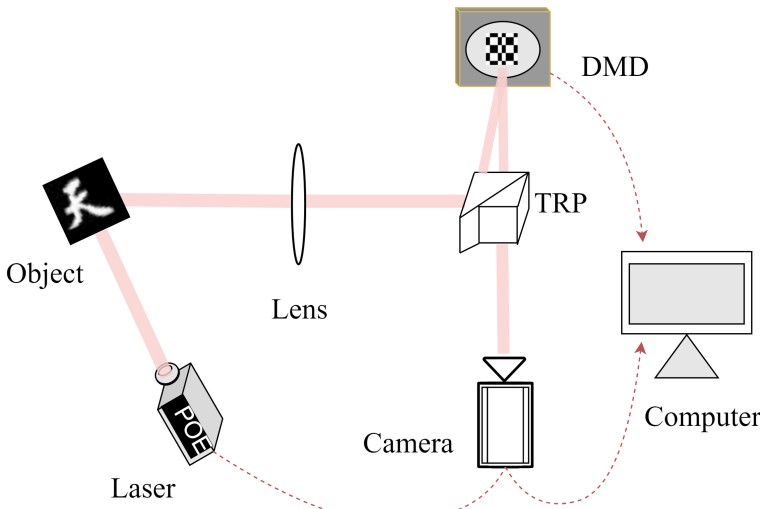

**Figure 4.** The schematic diagrams of CGI based on a ToF camera.

### 3.1. Recovering Depth Images Using Different Methods

This section provides the depth image recovery using our scheme of Figure 4 in indoor conditions with ambient light, where a binary object of a Chinese character of the sky is used as the reconstruction target shown in Figure 5a. For comparisons, a traditional ToF depth map is first supplied in Figure 5b, which has an obvious noisy background and lower contrast compared with that of Figure 5a. In CGI experiments, the bucket signals are acquired by summing the intensity of all the pixels of the area of interest of the ToF sensor regardless of what algorithms we refer to, where they are synchronized with the Hadamard bases that are projected on the DMD. In the framework, three well-used image reconstruction algorithms including CGI, basis pursuit (BP) and total variation augmented Lagrangian Alternating Direction Algorithm (TVAL3) are used. It is shown in Figure 5c that the images by CGI are reconstructed at the low SR of 6.25%, 12.5%, 18.75%, 25%, 31.25% and 37.5%, respectively. It is obvious that the image quality increases with the increase in the SR as expected. Therefore, it is believed empirically that the object images could also be successfully recovered by CS-based reconstruction algorithms of BP and TVAL3, which are shown in Figure 5d,e. Unfortunately, the images by BP do not have a better reconstruction quality compared with CGI because of much less SR. However, intuitively, the images reconstructed by TVAL3 appear to have better quality compared to those reconstructed by CGI at the same SR.

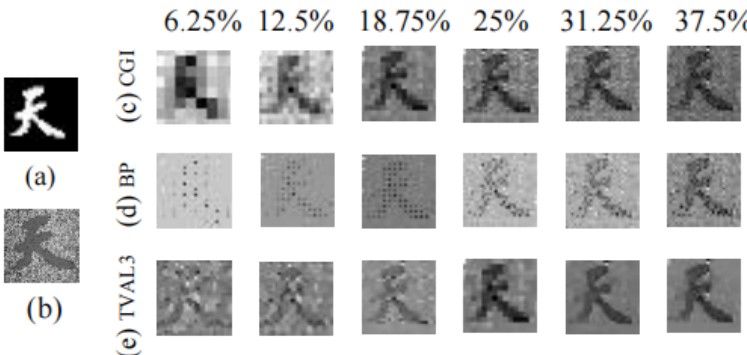

**Figure 5.** Experimental results of imaging reconstruction using depth maps at different SRs. (**a**) target object, (**b**) ToF image, (**c**–**e**) the recovered images by CGI, BP and TVAL3. The SRs from left to right is 6.25%, 12.5%, 18.75%, 25%, 31.25% and 37.5%.

In accordance with the principles of ToF cameras, we can ascertain a linear relationship between phase and depth. Consequently, we conduct histogram analysis of the phase map obtained from the camera, aiming to analyze the sources of noise and the reasons for negative depth values. Through our research, we have determined that the signals collected by the detection array primarily consist of three components: photons reflected back from the target object, environmental noise, and photons resulting from multi-path reflections. Figure 6a represents the original depth image obtained using a ToF camera. To more directly illustrate the reasons for the negative aspects of the image, we use the grayscale representation in Figure 6b to depict the original depth map. We configured the camera with an initial frequency of 60 Hz, and based on the camera's internal parameters, the maximum measurable distance is limited to 5 m. In regions where the sensor receives photons reflected from multiple paths, and conversely, in areas where the sensor does not detect any photons, the camera defaults to measuring the phase values as the maximum possible. This is exemplified in Figure 6a, where the region with a depth value Z of 4.978 is highlighted. In contrast, the region with a depth value of 2.006 corresponds to photons reflected back from objects. It is evident that the values in the regions where photons are reflected back from the object are smaller than in other areas. This also explains why the resulting image displays a negative depth image.

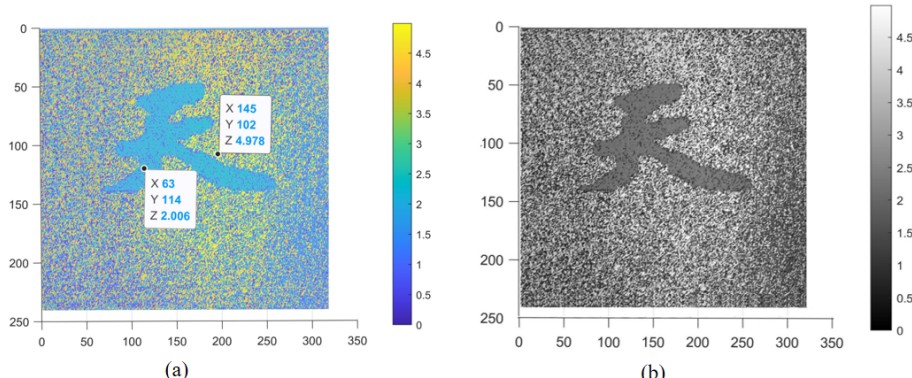

(a)

(b)

**Figure 6.** The original images exported by the ToF camera. (**a**) ToF original depth image, and (**b**) the original depth image in grayscale.

In order to quantitatively evaluate the image quality, the *PSNR* in decibel (dB) is introduced as

$$PSNR = 10 \cdot \log_{10}\left(\frac{MAX^2}{MSE}\right),$$
(13)

where $MAX = 2^k - 1$, which is the maximum of the processed image determined by the image bit depth $k$ and the MSE that is used to measure the difference between the reconstructed image $\widetilde{O}$ and the ground truth image $O$ is given by

$$MSE = \frac{1}{mn} \sum_{i=0}^{m-1} \sum_{j=0}^{n-1} (\widetilde{O}(i,j) - O(i,j))^2. \tag{14}$$

Here, $(i,j)(i = 0, 1, 2, \ldots, (m-1); j = 0, 1, 2, \ldots, (n-1))$ is the pixel coordinates of the image.

The relationship between the PSNRs of reconstructed images using different reconstruction algorithms and the SR is compared, which is shown in Figure 7. It is obvious that the PSNR by each reconstructed algorithm all increases with the increase in the SR while the image by TVAL3 has the highest PSNR at each given SR. For example, when the sampling rate is 12.5%, the PSNR values of three algorithms are 8.23 dB, 7.64 dB and 8.52 dB, respectively. Compared to the original depth map with a PSNR value of 5.70 dB, the image quality has improved by 1.44 times, 1.34 times, and 1.49 times, respectively. When the sampling rate is 25%, the PSNR values of the three algorithms are 9.44 dB, 9.76 dB and 9.82 dB, respectively. Compared to the original depth map, the image quality has improved by 1.66 times, 1.71 times, and 1.72 times, respectively. Therefore, by combining the CGI-based scheme and the ToF camera, the noise that most impacts the image quality due to ambient light and the defects in the detector could be suppressed to achieve higher-quality depth images.

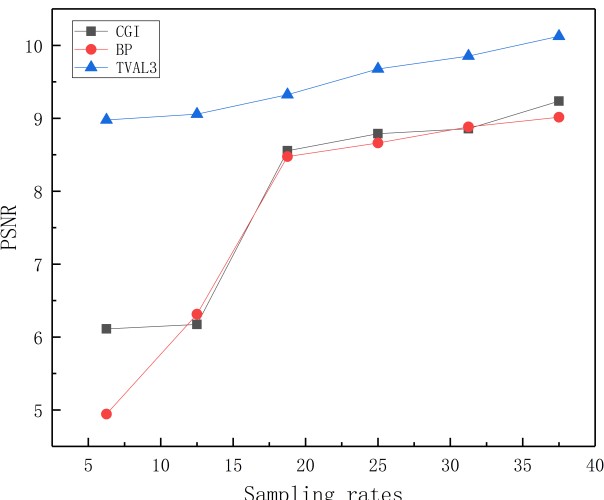

**Figure 7.** Plots of the PSNRs of the reconstructed depth images versus the SRs by different algorithms. The black, red and blue lines denote the PSNRs by CGI, BP and TVAL3.

In Figure 6, we have identified certain issues in the images, including multipath interference and noise. Since the effective information of the target object is within a range of less than 3 m, we have incorporated a low-pass filter into the later stages of the image processing. This filter helps by nullifying any information exceeding the 3 m threshold. Within this framework, we have opted for four commonly employed image reconstruction algorithms for the purpose of image recovery, namely CGI, BP, TVAL3, and an untrained DL network. It is shown in Figure 8b that the images are reconstructed at the low SR of 6.25%, 12.5%, 18.75%, 25%, 31.25 and 37.5%. It is obvious that the image quality increases with the increase in the SR as expected. Therefore, it is believed empirically that the object images could also be successfully recovered by CS-based reconstruction algorithms of BP and TVAL3, which are shown in Figure 8c,d. Unfortunately, the images by BP do not have a better reconstruction quality compared with CGI because of much less SR. However, intuitively, TVAL3 can reconstruct almost as good even better images as CGI at the same

SR. Fortunately, the untrained DL-based method can recover the target images with much better SNR compared with the other three methods, which is depicted in Figure 8e. Note that the batch size, learning rate and SR are set to be 1, 0.9 and 0.25, respectively. The best-reconstruction image is obtained while the iteration number of DL is chosen to be 300.

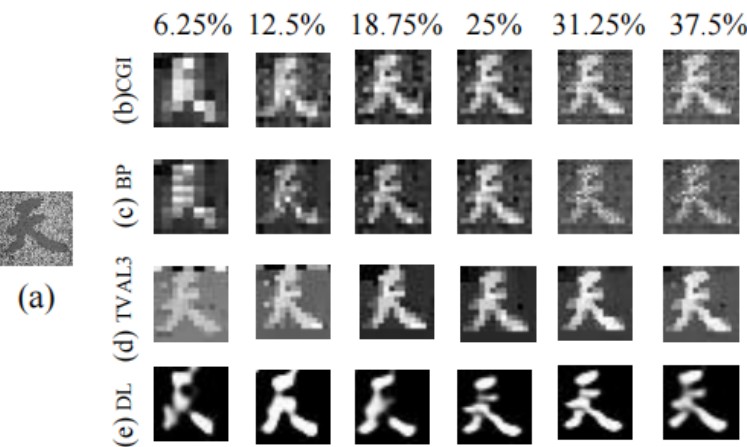

**Figure 8.** Experimental results of imaging reconstruction at different SRs. (**a**) ToF image, (**b**–**e**) the recovered images by CGI, BP, TVAL3 and DL. The SRs from left to right are 6.25%, 12.5%, 18.75%, 25%, 31.25 and 37.5%.

The relationship between the PSNRs of reconstructed images using different reconstruction algorithms and the SR is compared, which is shown in Figure 9. It is obvious that the PSNR by each reconstructed algorithm increases with the increase in the SR while the image by DL has the highest PSNR at each given SR. The PSNR by DL is 11.42 dB at even 6.25%, which is much higher than those by BP, CGI and TVAL3. However, in comparison to the original ToF images with a PSNR of 5.70 dB, the image quality was notably enhanced when utilizing the framework introduced in this paper at a sampling rate of 6.25 for four distinct reconstruction algorithms. The improvements in image quality were as follows: 1.35 times, 1.08 times, 1.47 times, and 1.75 times, respectively. When the sampling rate reached 25%, the PSNR of CGI-reconstructed images increased to 9.44 dB, translating to a 1.66-fold enhancement in image quality over the original ToF images. BP-reconstructed images achieved a PSNR of 9.76 dB, equating to a 1.71-fold improvement in image quality compared to the original ToF images. TVAL3-reconstructed images obtained a PSNR of 9.82 dB, indicating a 1.72-fold increase in image quality relative to the original ToF images. DL-reconstructed images exhibited a remarkable PSNR of 11.39 dB, signifying a twofold improvement in image quality compared to the original ToF images. However, only the images by DL can be clearly recovered with high quality at not only the much lower SR of 6.25%, but also other higher SRs, while those by other algorithms are either almost blurred, or still have a lot of noisy background. Thus, it could be deduced that the optimization effect of the DL algorithm in depth image reconstruction is more pronounced as the other algorithms' performances in depth image reconstruction are not so well as those in intensity image reconstruction while the DL algorithm performs even better in depth image reconstruction. These findings demonstrate that the DL algorithm produces superior image quality in the reconstruction of detailed maps. It has to be admitted that the filtering in DL image reconstruction also plays a great role in suppressing noise, some of which is inevitable for a ToF camera thanks to the multi-path interference and the mixed pixels and so on could be eliminated.

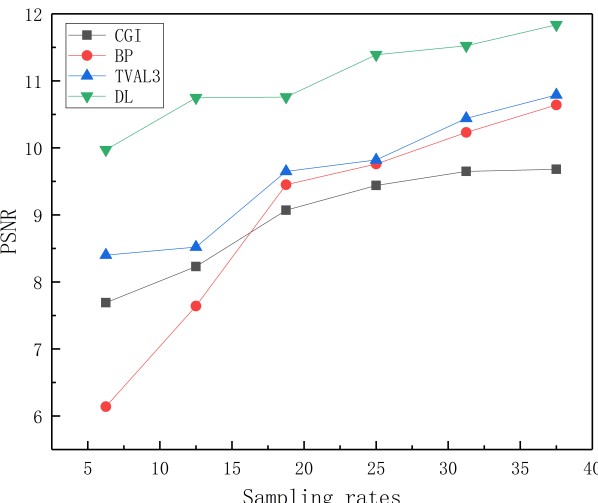

**Figure 9.** Plots comparing the PSNR and SRs for depth image reconstruction using different algorithms after applying low-pass filtering.

### 3.2. Recovering Depth Images Using Different Methods through Scattering Media

Due to the influence of scattering media, the image quality of ToF cameras deteriorates when imaging through such media. However, the CGI possesses strong anti-interference capabilities and can effectively resist the impact of atmospheric turbulence and scattering media on the image quality. To test whether the proposed scheme can image an object through a scattering medium, we inserted a 0.5 mm thickness transparent plastic sheet into the optical path between the ToF camera and the TRP in Figure 4 as the scattering medium, where the sheet is covered with a layer of dust particles of various sizes to scatter the light beam illuminated on it. In this scenario, the data collected by the ToF camera can be categorized into information from objects affected by scattering media and information from objects unaffected by scattering media. The imaging of the target object information in the ToF camera can be represented as

$$D(x) = \alpha D(x) + D^s(x), \tag{15}$$

In this context, $\alpha(0 < \alpha < 1)$ represents the transmittance ratio of the scattering medium, while $\alpha D(x)$ and $D^s(x)$, respectively, represent the distribution of information regarding reflected light and scattered light from the object. The other experimental conditions and the setting of the ToF camera are not changed and the experimental results through the scattering medium are shown in Figure 10, while the corresponding PSNR values are displayed in Figure 11. In Figure 10a, the ToF depth map is almost completely fuzzy and indistinguishable, which means that the traditional ToF camera does not work in this situation. However, by using the ToF camera as a detector in the scheme, which could preserve more high-frequency information of the ToF maps that determine how sharp the edges of the maps are, the depth images can be successfully reconstructed by CGI, BP and TVAL3 at the SRs of 6.25%, 12.5%, 18.75%, 25%, 31.25% and 37.5%, which are shown in Figure 10b–d, respectively. By comparison with the reconstruction images in Figures 7 and 11, it is found that the PSNR of each recovered image by each algorithms at each given SR becomes a little bit worse, at least in most cases we have measured. For example, when the sampling rate is 12.5%, the PSNR values of three algorithms are 5.09 dB, 4.91 dB and 6.17 dB. When the sampling rate is 25%, the PSNR values of the three algorithms are 6.16 dB, 6.05 dB and 6.81 dB. Although image quality recovery in a scattering medium is lower than in standard environments, our proposed solution still facilitates the use of ToF cameras in scattering media, offering a fresh approach to employing ToF cameras in complex environments.

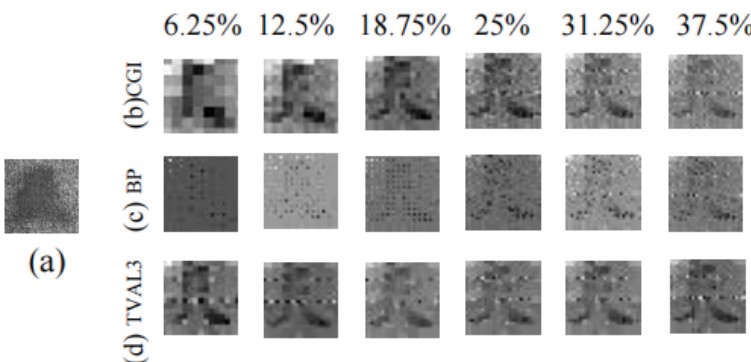

**Figure 10.** Experimental results of reconstruction using the depth maps through the scattering media at different SRs. (**a**) ToF image, (**b**–**d**) the recovered images by CGI, BP and TVAL3. The SRs from left to right are 6.25%, 12.5%, 18.75%, 25%, 31.25% and 37.5%.

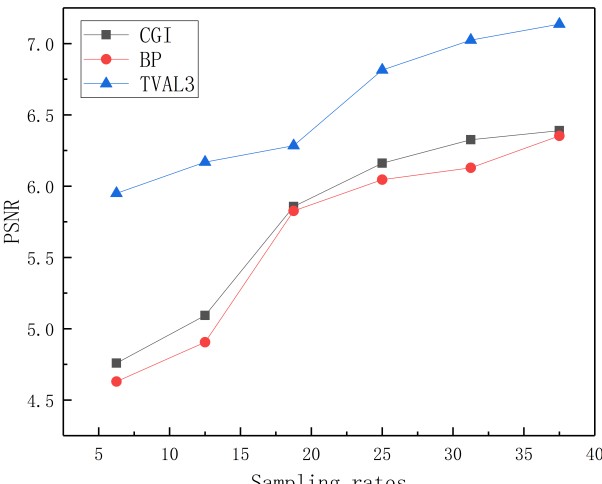

**Figure 11.** Plots comparing the PSNR and SRs for the reconstruction of depth images through scattering media using different algorithms.

Similarly, when imaging through scattering media using ToF cameras, we incorporate a low-pass filter in our post-processing image treatment. We still employ four image reconstruction algorithms for image restoration, namely CGI, BP, TVAL3, and untrained DL network. It is shown in Figure 12b–e that the images are reconstructed at the low SR of 6.25%, 12.5%, 18.75%, 25%, 31.25 and 37.5%. As expected, when using a low-pass filter, the quality of Image 12 is superior to Image 10, and the image quality increases with the increase in SR. Notably, when reconstructing images through scattering media at a low SR of 6.25% in Figure 13, the PSNRs of depth maps reconstructed by CGI, BP, TVAL3, and DL are 7.03, 7.46, 7.52 and 9.01 dB, respectively. When reconstructing images through scattering media at SR of 25%, the PSNRs of depth images reconstructed by CGI, BP, TVAL3, and DL are 9.59, 9.84, 10.21 and 11.73 dB, respectively. Correspondingly, the PSNRs of the reconstructed depth images are 4.76, 5.95, and 9.00 dB, respectively. These results suggest that DL is particularly effective in reconstructing high-quality images through scattering media at low SRs. DL needs to input an image into the network to be the original image used for iteration, and the number of the iteration and the SR that need to obtain the clear recovered image will decrease when the image is blurry. But this explanation still needs more experimental verification. Therefore, by combing the CGI-based scheme and the ToF camera, the interference caused by the scattering media could be suppressed to realize the imaging of the target object successfully.

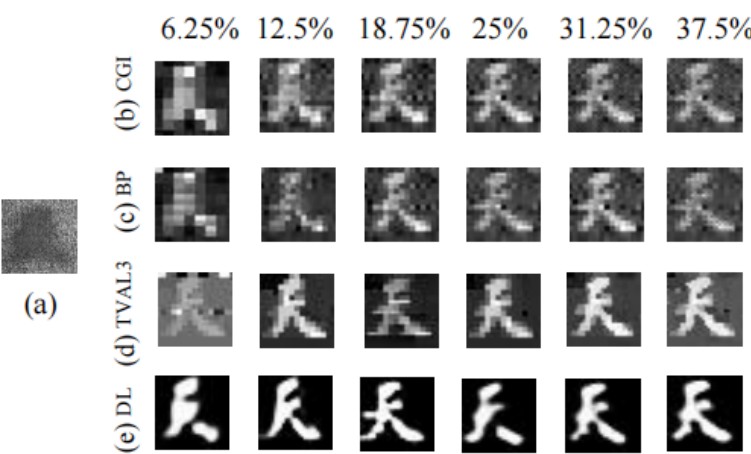

**Figure 12.** Experimental results of reconstruction using the depth maps through the scattering media at different SRs. (**a**) ToF image, (**b**–**d**) the recovered images by CGI, BP and TVAL3. The SRs from left to right are 6.25%, 12.5%, 18.75%, 25%, 31.25% and 37.5%.

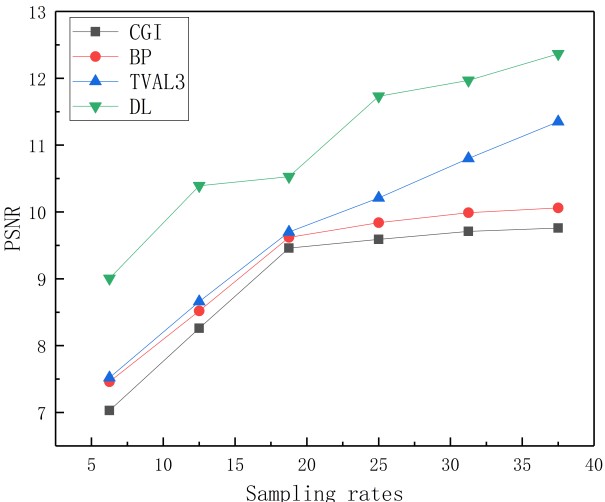

**Figure 13.** Plots comparing PSNR and SRs for the reconstruction of depth images through scattering media using different algorithms after applying low-pass filtering.

## 4. Conclusions

In conclusion, we have effectively demonstrated innovative applications of ToF cameras as bucket-detectors in both typical environments and scattering media. The introduction of the second-order correlation transformation efficiently mitigates noise originating from the ToF camera itself and direct ambient light. Our approach capitalizes on ToF camera depth information, utilizing various algorithms, including CGI, CS, and DL, to reconstruct images of target objects. Within our proposed framework, we address the common challenges of low resolution and a low signal-to-noise ratio encountered in depth cameras. Research findings indicate that both CGI and CS algorithms are capable of reconstructing negative depth images, and we have meticulously analyzed and mitigated the factors causing negative depth images. Particularly at low sampling rates, CGI and CS yield higher-quality image recovery compared to the original depth images. Remarkably, untrained deep learning networks exhibit substantial advantages in super-low subsampling, achieving a sampling rate of 6.25%, well below the Nyquist limit for image recovery. Furthermore, we have successfully facilitated the deployment of ToF cameras in scattering media, with the proof-of-concept demonstration underscoring the potential of ToF cameras in challenging scenarios such as haze, rain, snow, and underwater environments. Our research provides another avenue for the application of ToF cameras and opens up the possibility of integrating ToF cameras into other imaging systems.

**Author Contributions:** Conceptualization, T.-L.W.; methodology, T.-L.W. and L.A.; validation, T.-L.W. and L.A.; writing—original draft preparation, T.-L.W. and L.A.; writing—review and editing, Z.-B.S.; data curation, T.-L.W., L.A. and J.Z.; supervision, Z.-B.S.; funding acquisition, Z.-B.S. All authors have read and agreed to the published version of the manuscript.

**Funding:** This study was funded by National key research and development program (2016YFE0131500); Scientific Instrument Developing Project of the Chinese Academy of Sciences, Grant No.YJKYYQ20190008.

**Institutional Review Board Statement:** Not applicable.

**Informed Consent Statement:** Not applicable.

**Data Availability Statement:** Research data from this study will be made available upon request by contacting the authors.

**Conflicts of Interest:** The authors declare no conflict of interest.

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
