# Peer review of "Reconstructing Depth Images for Time-of-Flight Cameras Based on Second-Order Correlation Functions"

_photonics, doi:10.3390/photonics10111223_

Round 1

Reviewer 1 Report

Comments and Suggestions for Authors

This manuscript is about depth image reconstruction using time-of-flight (ToF) camera in noisy environment. Authors claim that combination of compressed-sensing and neural network can be used to reconstruct the depth image properly even with a very low spatial sampling rate. Although interesting, the design of experiment seems to lack scientific rigor, and major revision will be needed before publishing.

<Major Issues>

-      It is suggested that authors divide the section 3 into a couple of subsection, because it is hard to follow otherwise. Figures 7, 9, and 11 are sharing the same figure captions, so it is tough to figure out what their differences are if no further description is added.

-      The whole experiment is using a 2-dimensional sample with a shape of a Chinese letter, which defeats the purpose of using ToF camera. For better quality research, a 3-D object with some volumetric feature is desirable.

<Minor Issues>

-      Schematics for optical setup can be improved, by replacing the arrow shapes with plane bold lines for ray propagation.

-      For scattering medium, only the thickness is specified. Its scattering property needs to be given, either as a reduced scattering coefficient or as an angle distribution of transmitted photons

-      Line 102 : … assumed as just shown on the …

-      Line 193 : ref number should be positioned before ‘for’

-      Eq 3 and Line 116 : better use the standard notation for convolution (asterisk)

-      Line 226, 272 : as good or even better …

-      Fig 6 caption : Explain what those two points on Fig 6(a) means

-      Fig 6 caption : is -> in

-      Line 275 : Noted -> Note

-      Line 312 : others -> other

Reviewer 2 Report

Comments and Suggestions for Authors

This paper proposed a method to reconstruct depth images with time-of-flight cameras and  second-order correlation method. The second order correlation method shows an effective result on reducing system noise and eliminating direct ambient light noise. This method is interesting and some comments are listed below:

1. Figure 2 is not very informative,such as the name of each component, the lens information, what the dash lines connected to the computer means, resolution of the detector, etc.  2. THe SR range in Figure5 is from 6.25-37.5%, what is the limit SR the TVAL3 can do, what is the main issue to limit the performance? 3. In this paper, the experiment is done on a pretty simple word grey level image. Did you try it on a more sophisticated pattern to check the resolution and accuracy?  

Reviewer 3 Report

Comments and Suggestions for Authors

Please find the comments referring to the paper as an attachment.

Round 2

Reviewer 1 Report

Comments and Suggestions for Authors

Good revision. No problems with publishing it.

Reviewer 3 Report

Comments and Suggestions for Authors

All comments of the reviewer have been included in the revised version of the paper. I recommend publication of this paper in its current form.